# A quasi-solid-state high-rate lithium sulfur positive electrode incorporating $Li_{10}GeP_2S_{12}$

Boyi Pang[1,2], Huanxin Li [1,2], Yiming Guo[2,3], Bochen Li[3], Feiran Li[4], Huw C. W. Parks[1,2], Liam R. Bird[2,5], Thomas S. Miller [1,2,3], Paul R. Shearing[2,5], Rhodri Jervis [1,2,3] & James B. Robinson [1,2,3] ✉

Lithium–sulfur batteries offer high theoretical energy density for advanced energy storage, but practical deployment is hindered by the polysulfide shuttle effect and sluggish kinetics in conventional catholytes. Here, we develop a high-rate sulfur cathode by integrating $Li_{10}GeP_2S_{12}$, a highly ion-conductive solid-state electrolyte, directly into the positive electrode. We systematically investigate the influence of solvent systems and binders on electrochemical performance, while optimising the slurry casting process. Electrochemical tests demonstrate that the addition of $Li_{10}GeP_2S_{12}$ improved lithium-ion transport, reduced internal resistance, and enhanced reaction kinetics, leading to a high initial capacity of over 1400 mAh g$^{-1}$. We observe high-capacity retention at high current densities (1 C) with the positive electrode exhibiting a stable capacity of 800 mAh g$^{-1}$, significantly outperforming control samples fabricated without $Li_{10}GeP_2S_{12}$. This study confirms that the integration of $Li_{10}GeP_2S_{12}$ into the positive electrode enhances the performance of quasi-solid-state lithium–sulfur batteries, offering potential for future improvements based on the optimisation of lithium-ion conducting pathways in the positive electrode

As global energy demand continues to grow, the need for efficient energy storage solutions has become more pressing, especially in the areas of renewable energy utilisation, portable electronic devices, electric vehicles (EVs) and grid storage systems. While lithium-ion (Li-ion) batteries are the dominant technology in these fields, continued improvement in the technology is limited by their energy density approaching the theoretical limit of graphite, restricting the devices' potential deployment in energy density-sensitive applications[1–3]. Of the next-generation battery technologies, Lithium sulfur (Li-S) offers substantial benefits in this area due to the superior theoretical gravimetric energy density (~2600 Wh kg$^{-1}$). In addition, there are a range of potential benefits of this technology when considering the cost, sustainability and safety of devices[4]. However, the commercialisation of Li-S batteries has been hampered by a series of issues in their practical application, resulting in realised energy densities and lifetimes that are much lower than the theoretical values[5]. Of these, the main challenge is the 'shuttle effect', where the intermediate lithium polysulfides ($Li_2S_x$, $4 \leq x \leq 8$) produced during discharge dissolve in the liquid electrolyte and migrate between electrodes. This leads to loss of active material, severe capacity decay and reduced Coulombic efficiency[6,7]. Since sulfur is an

insulator, poor electrical conductivity, low active material utilisation and poor rate performance of the battery have also been observed[8]. Furthermore, the complex reaction mechanism of the Li-S conversion chemistry has generally resulted in poor rate performance when compared with current Li-ion technology which are based on intercalation storage mechanisms.

In order to address the limitations of conventional Li-S batteries, researchers have developed a range of strategies which can be deployed at the positive electrode, such as encapsulating sulfur in a conductive porous body[9,10], developing new separators using unique diaphragms[11], and incorporating intercalation layers, electrolyte additives[12], hybrid negative electrode structures[13,14], and novel binders[15–17]. In recent years, there has been an increased focus on approaches that alter the operating mechanism of the cell to mitigate the issues observed. These include solid-state approaches which use Li-ion conducting solid electrolytes or gelled structures that permit transport of Li-ions from the negative electrode to the positive electrode, but prevent polysulfide shuttle[18], with innovations enabled by improvements in the ionic conductivity of the solid and gel layers within cells[19–21]. Further developments have focused on deploying a liquid electrolyte that has limited polysulfide solubility, driving the sulfur

[1]Advanced Propulsion Lab (APL), University College London, Marshgate, London, E20 2AE, UK. [2]Faraday Institution, Quad One, Becquerel Avenue, Harwell Science and Innovation Campus, Didcot, OX11 0RA, UK. [3]Electrochemial Innovation Lab (EIL), University College London, Gower Street, London, WC1E 7JE, UK. [4]Imperial College London, South Kensington Campus, London, SW7 2AZ, UK. [5]ZERO Institute, University of Oxford, Holywell House, Osney Mead, OX2 0ES Oxford, UK. ✉e-mail: j.b.robinson@ucl.ac.uk

reduction towards a solid-state type of mechanism, whereby the sulfur conversion pathway is not entirely dependent on the formation and dissolution of liquid intermediates. These so-called quasi-solid-state (QSS) Li-S batteries offer compelling benefits in the manufacturing process, enabling cells to be filled in a similar manner to Li-ion technology. Crucially, this approach also significantly reduces the electrolyte volume, thereby favourably affecting the practically realisable gravimetric energy density in cells. In addition, as the dissolution of polysulfides in the electrolyte is limited, adverse effects on viscosity which occur due to dissolution polysulfides and resulting resistance increase in conventional Li-S systems are greatly reduced or even eliminated[12,22–24].

Amongst the solid electrolytes which have been explored in the literature, sulfide materials stand out due to their high ionic conductivity, low crystal boundary resistance, compatible interface with sulfur-based positive electrodes, and ease of processing. Here, $Li_{10}GeP_2S_{12}$ (LGPS) is perhaps the most widely explored due to an extremely high ionic conductivity of $1.2 \times 10^{-2}$ S cm$^{-1}$ at room temperature[25]. The use of an LGPS solid electrolyte mitigates the shuttle effect, and its high Li-ion conductivity allows for rapid Li-ion migration, lowering the internal impedance of the cell. Alongside this, the Li-ion transport matches the reaction rate of the sulfur electrode, improving the efficiency of sulfur conversion in the cell[26–28]. Despite this promise, LGPS is thermodynamically unstable with Li metal[26], undergoing a reduction reaction with the Li metal negative electrode to form a hybrid ion/electron conducting $Li_2S$-$Li_3P$-$Li_xGe$ interface[29], which promotes the growth of lithium dendrites[30,31]. This reaction continues as long as the LGPS remains in contact with lithium metal, leading to LGPS degradation and growth of the interfacial layer, resulting in larger interfacial resistances, capacity fade, and ultimately short-circuit[31,32]. A range of approaches has been adopted, including using an intermediate layer[26], surface modification or passivation of lithium metal electrodes[32,33], alloying lithium metal[34,35], employing a bilayered solid-state electrolyte configuration[23,24], and incorporating ionic liquids into the LGPS[27,36]. However, there are few studies on the direct incorporation of solid-state electrolytes into the positive electrode slurry of lithium–sulfur batteries[37,38]. By incorporating LGPS directly in the positive electrode, the Li-ion conductivity and electrochemical performance will be improved, and the side reactions caused by direct contact between LGPS and lithium metal will be avoided. In this paper, we present a method for the preparation of sulfur composite positive electrodes for quasi-solid-state lithium–sulfur batteries. The main component of the sulfur composite electrode consists of a sulfur/carbon composite, with LGPS added as a solid-electrolyte additive and poly(vinylidene fluoride-co-hexafluoropropylene) (PVDF-HFP) used as a binder alongside a small mass of super P carbon (SP) to enhance the conductivity of the electrode. A range of solvents is systematically investigated to explore routes to scalable development, with optimisations undertaken for both the binder and coating process. The electrochemical performance of coin cells assembled from this positive electrode was tested and shows excellent capacity and rate performance. Finally, to investigate further how LGPS changes the reaction kinetics of Li-S batteries and affects battery performance, a range of physical and electrochemical characterisation was conducted. The results demonstrate another pathway to improving the performance of Li-S cells in a quasi-solid-state mechanism, offering opportunities to realise full cell energy densities which exceed the theoretical limits of Li-ion in the future.

## Results and discussion
### Characterisation of LGPS behaviour
To explore the impact of solvent choice on slurry stability, LGPS was immersed as described in six different solutions: n-methyl-2-pyrrolidone (NMP), dihydrolevoglucosenone (Cyrene), N,N-dimethylformamide (DMF), dimethyl sulfoxide (DMSO), ethylene acetate (EA) and hexyl butyrate (HB), as described in the methods section. Before immersion, LGPS was observed to be an off-white, fine powder, with this changing following evaporation of the solvents, indicating reactivity. When immersing the LGPS in some solvents such as NMP, Cyrene, DMF, and DMSO, permanent discoloration of the LGPS results (images shown in Fig. S1),

which indicates an adverse reaction. However, two solvents, EA and HB, showed no significant change in colour or hardness after immersion in the solvent.

The commonly used solvents NMP, cyrene, DMF, and DMSO are polar solvents, and EA and HB are non-polar solvents[39]. Combined with the results of the immersion experiments, it was hypothesised that LGPS might react with polar solvents but not with non-polar solvents. The solvents cyrene, DMF, and DMSO were excluded because they underwent a change in properties visible to the naked eye. To further investigate whether LGPS would react with solvents (NMP, EA & HB) that did not undergo significant colour changes, electrochemical impedance spectroscopy (EIS) tests were performed as shown in Fig. 1a–c. It can be seen that the impedance of the LGPS dissolved in NMP and then recrystallised is significantly higher, with an increase in impedance by a factor of about $10^2$. In contrast, the LGPS test impedance did not change significantly by orders of magnitude with and without the EA&HB treatment. Due to experimental challenges in accurately controlling the pressure and thickness pressed into the LGPS during the testing process, because the density, hardness, and degree of densification of the LGPS may change during the testing process, it is challenging to firmly conclude whether the LGPS is impacted by the treatment of EA&HB from the EIS results alone.

Further characterisation was conducted using X-ray diffraction (XRD) to assess whether the crystal structure of LGPS had changed because of the dissolution and recrystallisation. The results shown in Fig. 1d demonstrate that the crystal structure of LGPS after EA&HB treatments remained stable, with the positions of the main diffraction peaks not changing significantly. The absence of significant peak displacement and the appearance of new peaks indicate that LGPS did not undergo phase transformation or decomposition into polysulfides or other crystalline compounds such as $Li_4GeS_4$, after EA and HB treatments. The decrease in peak intensity may result from a reduction in crystallinity, indicating that the LGPS particles become more disordered after solvent treatment while preserving their original crystalline phase. It is possible that there is a process of dissolution and recrystallisation. Moreover, during the solvent treatment of LGPS, a slurry-like morphology was observed with the naked eye, which stuck to the bottom of the container like a scale after evaporation.

To verify that LGPS undergoes a dissolution-recrystallisation process after EA&HB treatment, a scanning electron microscope (SEM) was used to observe the microstructure, as shown in Fig. 1e, f, h, i. The original LGPS [Fig. 1e, h] has smaller, sharper particles with distinct edges, exhibiting typical crystalline particle aggregation characteristics. After EA&HB treatment, the LGPS particles become rounder and blunter overall, with a rough surface and blurred boundaries between particles, as shown in Fig. 1f. At lower magnification in Fig. 1i, large flat areas and crystalline-like structures are clearly visible, which may result from the regrowth and reorganisation of crystals. Meanwhile, the morphology of LGPS after NMP treatment [Fig. 1g, j] is completely different, having lost its original crystalline structure.

Overall, LGPS treated with EA and HB retains its characteristic crystalline phase, suggesting EA and HB can be used as solvents. Currently, there are few studies on the reaction of various solvents on LGPS, and further electrochemical tests are needed to confirm their effects on the cell performance[39–41].

Polyvinylidene fluoride (PVDF), a commonly used binder in Li-S battery positive electrodes, exhibits poor solubility in EA, which is the solvent of choice in this study. Consequently, we utilised PVDF-HFP as an alternative binder. PVDF-HFP introduces hexafluoropropylene (HFP) units to PVDF, a structural modification that disrupts the crystalline regions of PVDF and gives it partially amorphous regions. Since the HFP unit has stronger polarity, PVDF-HFP has better wettability to the electrolyte, and HFP easily forms local ionic conductive channels, which can improve the ionic conductivity in the electrode to some extent. The better polarity also means that the F-group is effective for polysulfides trapping/polysulfides shuttle reduction too[42]. However, its synthesis process is much more complicated and costly than PVDF, and its bonding ability is weaker, so it has not been widely used in the battery industry[43–45], with its use

primarily limited to the production of gel electrolytes[38]. Due to the high vapour pressure (10 kPa at 20 °C) and low boiling point (77 °C) of EA, rapid evaporation of the solvent occurs during the drying process, leading to binder migration and challenges in forming a homogeneous, dense coating

with strong adhesion (Fig. S2). To address this, HB, a solvent with a significantly higher boiling point (vapour pressure: 0.031 kPa at 20 °C, boiling point: 205 °C), was introduced as a co-solvent to improve manufacturability.

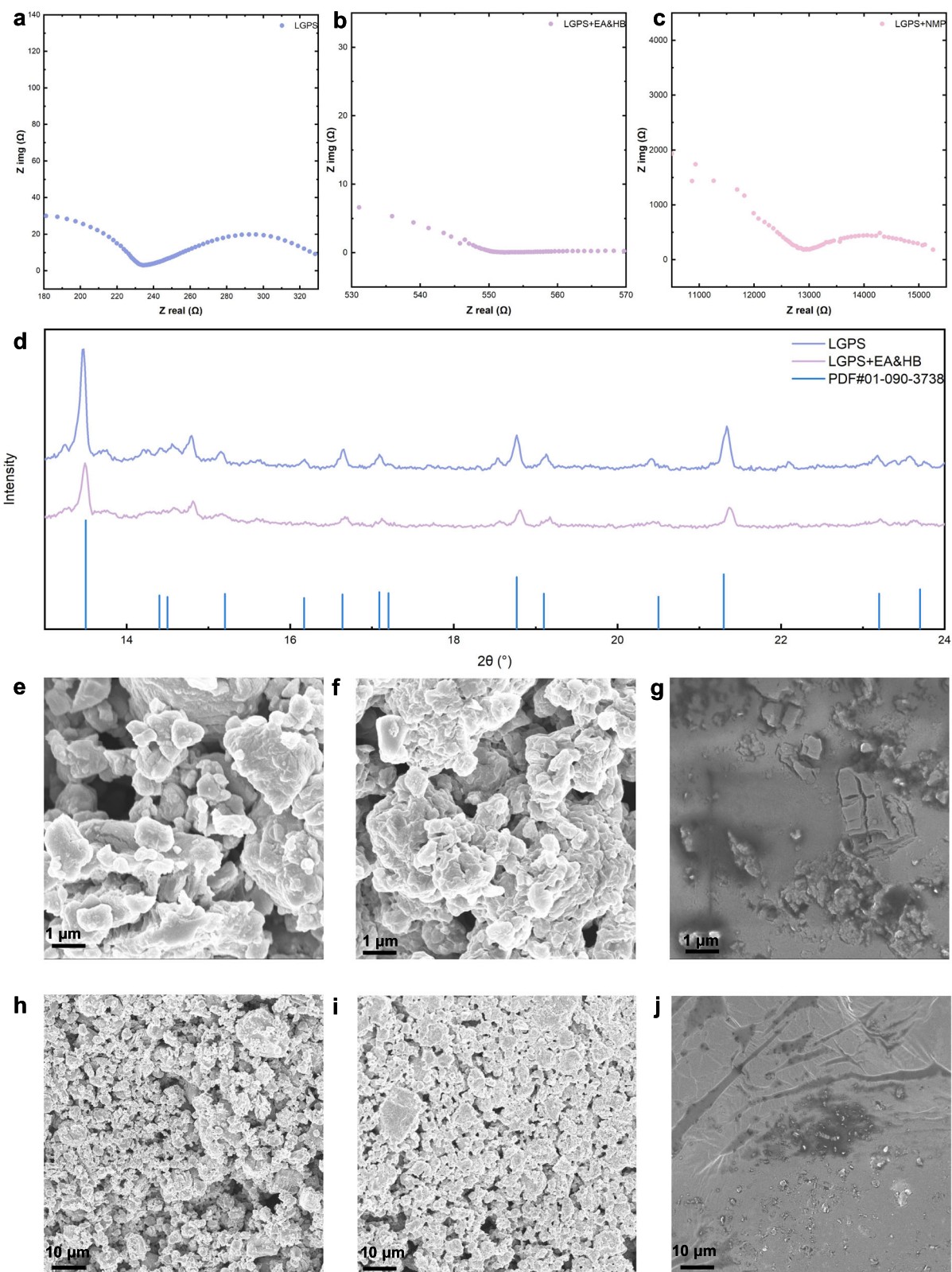

**Fig. 1 | Characterisation of LGPS with different solvents.** Electrochemical impedance spectroscopy (**a–c**), X-ray diffraction (**d**) and scanning electron microscope (**e–j**) of LGPS with different solvents: **a**, **e**, and **h** LGPS; **b**, **f**, and **i** LGPS with EA&HB; **c**, **g**, and **j** LGPS with NMP; **d** X-ray diffraction of LGPS, LGPS + EA&HB and LGPS reference peaks.

**Fig. 2 | Microstructure of the electrode.** Scanning electron microscope (**a**–**e**), optical microscopy measurements (**f**, **g**) and X-ray computed tomography (**h**, **i**) of the positive electrode incorporating sulfur, LGPS, ketjen black, and PVDF-HFP binder. **a** KB powder; **b** a side profile of the electrode highlighting the strong adhesion to the Al foil and dispersion of sulfur in the electrode; **c** the binder connectivity between the particles in the electrode; **d** KB with uniform particles in the electrode; **e** PVDF-HFP between cracks; **f** surface topography of the electrode obtained using optical microscopy; **g** cross-surface profile of the electrode highlighting the uniform thickness observed following manufacture; h) morphology of electrode; **i** morphology of S distribution within electrode.

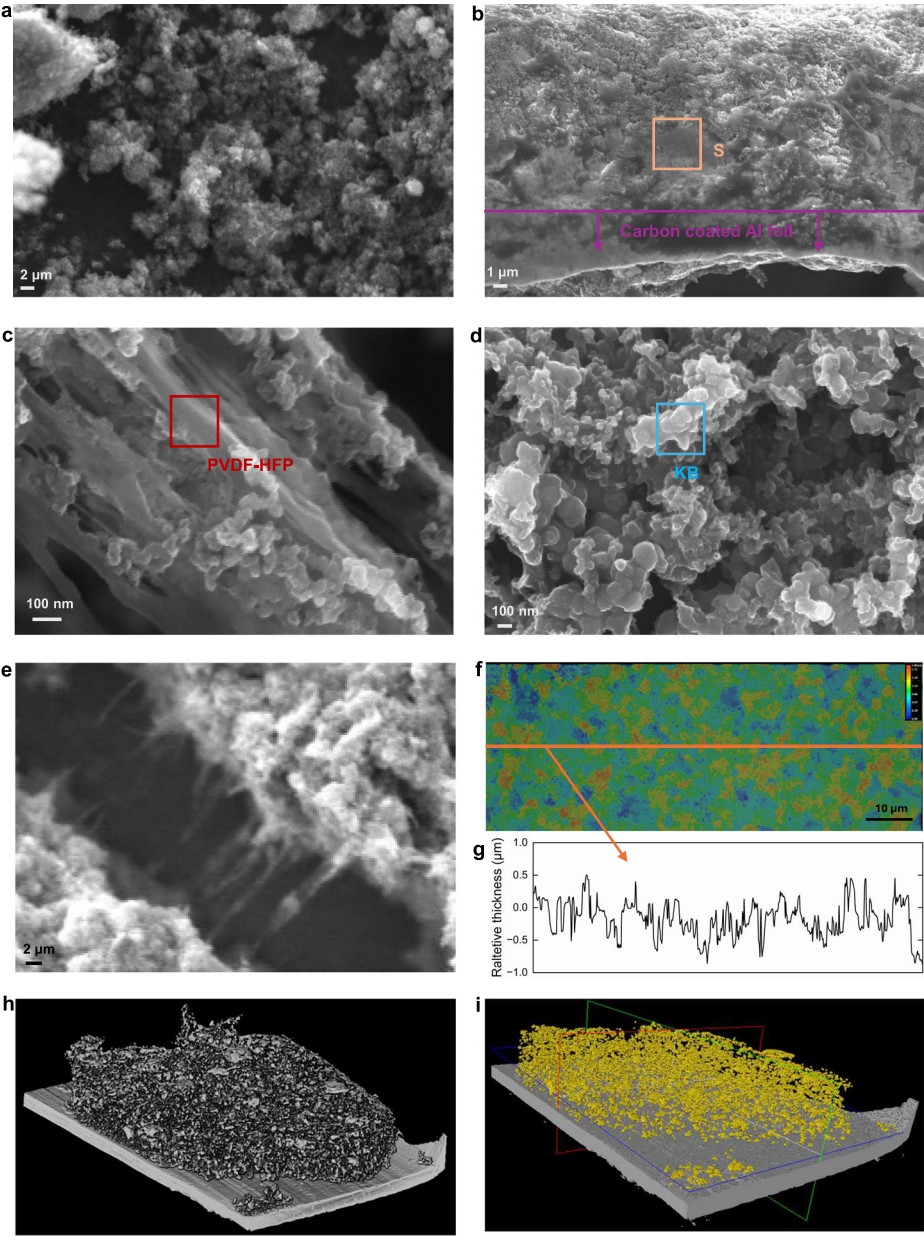

Initially, PVDF-HFP was dissolved in EA at 60 °C with continuous stirring for 30 min, but its solubility in HB was limited. Even in a heated EA/HB mixture at a 1:1 volume ratio, PVDF-HFP remained difficult to dissolve. Furthermore, upon introducing the PVDF-HFP solution into cold HB, it precipitated as a jelly-like colloid, which was clearly visible after coating (Fig. S3). After several trials, the optimised process was determined, as outlined in the experimental methodology. The resulting coatings appeared dense and free of colloidal residues, maintaining a powder-free surface even after folding and bending (Fig. S4). The SEM results shown in Fig. 2a–d illustrate that the binder PVDF-HFP was well dispersed and mixed in the positive electrode. The ketjen black (KB) particles (marked with a blue box) shown in the Fig. 2d are interconnected porous carbon structures. sulfur particles may be dispersed in the KB network as shown in Fig. 2h, i. Many pores can also be seen, which provide space for the volume change of sulfur during charging and discharging processes. Although cracks are inevitable in the cathode, Fig. 2e shows the binder between the cracks, which stabilises the structure while also providing a good conductive network. The digital microscope image [Fig. 2f, g] shows a uniform thickness across the surface of the positive electrode. This dispersion can also be seen to extend throughout wider regions of the electrode in the X-ray computed

tomography (XCT) data shown in Fig. 2h, i, indicating the electrode slurry was well-mixed and deposited in an appropriate fashion prior to testing.

Figure 3a, b, c, d shows that in the KB cathode material, C is uniformly distributed over the electrode except for a few voids while S is also uniformly distributed over a wide range, indicating that good distribution of S on the C substrate is conducive to the formation of an effective conductive network, and a small amount of F is detected in Fig. 3b, as being derived from the binder PVDF-HFP. The uniform distribution of S and C is still seen in Fig. 3g, h, and the distribution of F [Fig. 3i] seems to appear in the regions with lower sulfur content, while the distribution of F is similar to the C distribution, which reflects the disadvantage of PVDF-HFP's tendency to agglomerate. It is worth mentioning that the uniform distribution of Ge and P [Fig. 3j, k] indicates that the LGPS is uniformly distributed in the electrode material, which is crucial for enhancing the ionic conductivity of the material and can effectively promote the conduction of lithium ions.

## Electrochemical performance

The electrochemical performance of KB, PVDF-HFP, S, SP positive electrode (referred to as KB) and KB, PVDF-HFP, S, SP, LGPS positive electrode

**Fig. 3 | Energy-dispersive X-ray spectroscopy (EDS) analysis of KB and KB-LGPS positive electrodes. a** EDS raw image of KB positive electrode; **b** EDS analysis of KB positive electrode; **c** EDS mapping image of C of KB positive electrode; **d** EDS mapping image of S of KB positive electrode; **e** EDS raw image of KB + LGPS positive electrode; **f** EDS analysis of KB + LGPS positive electrode; **g** EDS mapping image of C of KB + LGPS positive electrode; **h** EDS mapping image of S of KB + LGPS positive electrode; **i** EDS mapping image of F of KB + LGPS positive electrode; **j** EDS mapping image of Ge of KB + LGPS positive electrode; **k** EDS mapping image of P of KB + LGPS positive electrode.

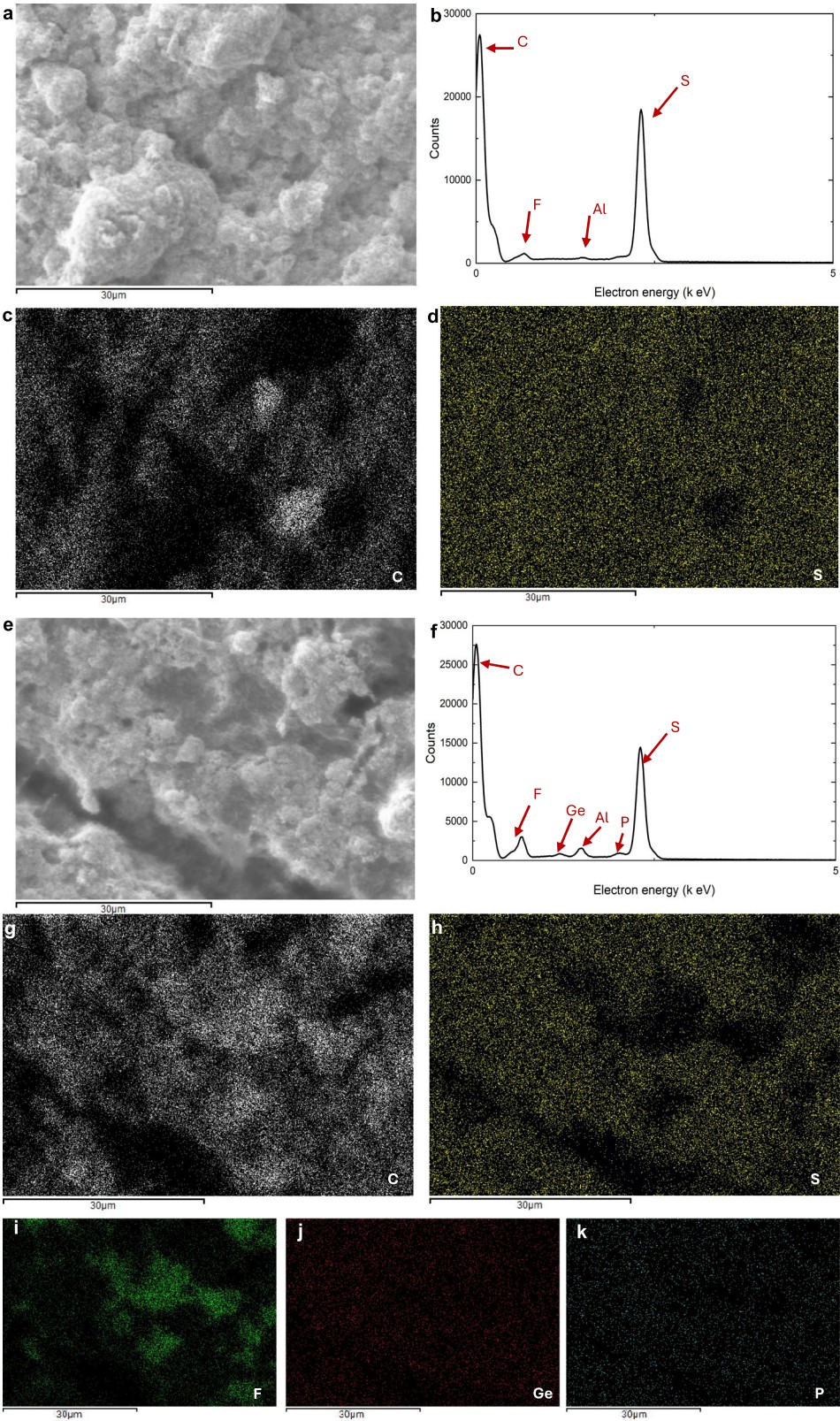

(referred to as KB-LGPS) was evaluated in a coin cell configuration. Figure 4a shows the charge-discharge curves of the two groups at 0.01 C and 1 C rates. Initially, a first cycle of 0.01 C was applied to both cells, with the KB-LGPS electrode achieving capacity of 1408 mAh g⁻¹, compared to 1265 mAh g⁻¹ for the KB control cell. The charge-discharge curves without LGPS showed overpotentials of 217 mV and 855 mV at 0.01 C and 1 C,

respectively. However, the charge-discharge curves with LGPS showed significantly lower overpotentials of 116 mV (0.01 C) and 464 mV (1 C). This demonstrates the enormous potential of LGPS in reducing overpotential. The discharge curve of KB-LGPS at 0.01 C exhibited two distinct plateaus, with a smoother transition between them compared to KB. This dual plateau behaviour has been previously observed in so-called quasi-

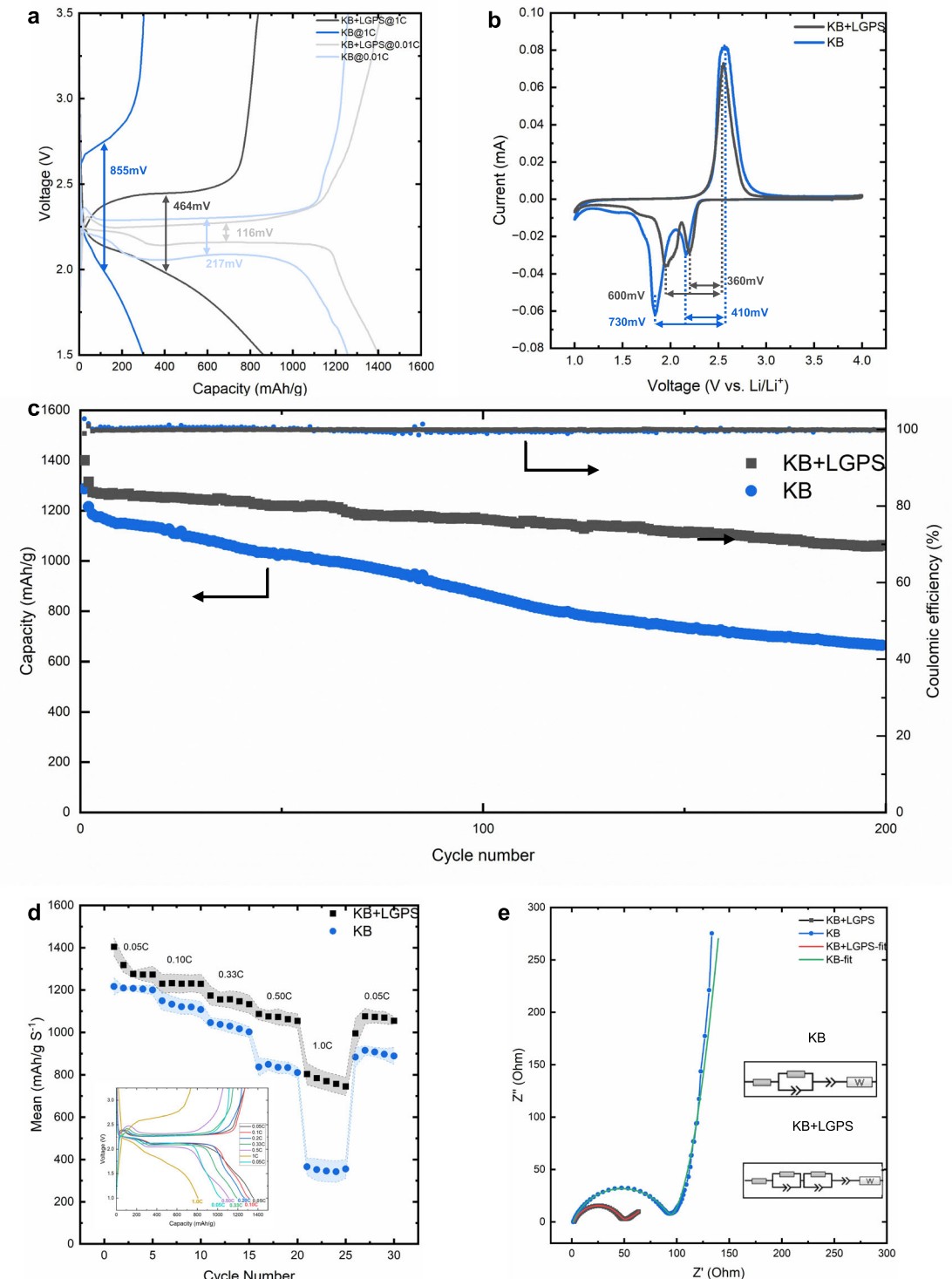

**Fig. 4 | Electrochemical performance of KB and KB-LGPS. a** charge and discharge at 0.01 C and 1.0 C; **b** 1st cycle CV at 0.01 mV s⁻¹; **c** long-term cycle performance the first cycle at 0.05 C then at 0.1 C; **d** mean rate performance with standard deviation shown in shadow and KB + LGPS discharge and charge curve; **e** EIS of fresh coin cells.

solid-state systems[46]. In addition, the relative lengths of the two plateaus, which are indicative of the discharging mechanism that occurs in a Li-S cell[47] can also be seen to be different in the two different systems, suggesting the addition of LGPS promotes a different reaction reduction pathway in the cell. To explore this in further detail, cyclic voltammetry of the cells was performed as shown in Fig. 4b. Both CV curves (Fig. 4b) show the typical redox reaction peaks found in lithium–sulfur batteries with an oxidation peak occurring around ~2.5 V associated with the oxidation of sulfur from lithium sulfide ($Li_2S$) to polysulfide and sulfur during charging. This peak

can be seen to occur at the same voltage for both the KB and KB-LGPS cells. The reduction peaks, associated with cell discharging, typically occur around 2.0 V and 2.3 V. The reduction peak at ca. 2.3 V corresponds to the reduction of polysulfides ($Li_2S_n$, $n > 2$) to longer-chain polysulfides (e.g., $Li_2S_4$, $Li_2S_6$). There is almost no significant difference between the peaks of the two curves at this position, indicating that the addition of LGPS does not change the reduction process of polysulfides toward longer-chain polysulfides significantly. The electrode with the addition of LGPS has a reduction peak near 2.0 V, which corresponds to the further reduction of

polysulfides to $Li_2S$. However, the second reduction peak without added LGPS can clearly be seen to shift to the negative direction near 1.8 V, indicating that the reaction requires a higher overpotential to drive it. Thus, LGPS improves the reaction kinetics during the reduction of polysulfides to $Li_2S$ and improves the electrical or ionic conductivity. Besides, the addition of LGPS reduced the peak potential difference of the electrode, indicating reduced electrode polarisation[48].

As shown in Fig. 4c, the KB + LGPS electrode exhibits significantly superior stability compared to the KB electrode during long-term cycling. After 200 cycles at a current density of 0.1 C, the capacity decay of the KB + LGPS electrode was only 0.09% per cycle, while the KB electrode rapidly decayed to just 56%. This indicates that the introduction of LGPS plays a positive role in extending cycle life. This may be attributed to the addition of LGPS, which enhances ionic conductivity in the cathode, facilitating the formation of more uniform ionic transport pathways and thereby reducing polarisation during cycling. Notably, both electrodes maintained a coulombic efficiency close to 100% throughout the cycling process, indicating good reversibility of the electrochemical reaction and effective suppression of side reactions. Nevertheless, the KB electrode exhibited more severe capacity decay, suggesting that the decay is not caused by side reactions but more likely due to structural degradation or deteriorated interface contact. The absence of LGPS results in discontinuous lithium-ion transport pathways in the KB electrode, leading to increased polarisation and eventual failure of active areas, thereby causing capacity decline.

To explore the impact of rate on the electrode performance, a series of increasing discharge currents was applied sequentially as shown in Fig. 4d. It can clearly be seen that the KB-LGPS electrode provides a slightly higher capacity at lower rates (0.05–0.33 C) with this difference remaining relatively constant. Combined with the long-cycle performance plots in Fig. 4c, we do not believe that the addition of LGPS significantly increases capacity at low rates. However, at 0.5 C, a significant difference emerged with this difference increasing to 40%, the KG-LGPS retaining a capacity of 1100 mAh $g^{-1}$ compared to the KB electrode's 900 mAh $g^{-1}$. When the discharge rate was further increased to 1 C the KB electrode retained only 400 mAh $g^{-1}$ in contrast to the KB-LGPS which maintained 800 mAh $g^{-1}$. Furthermore, the method of adding lithium-ion-conductive additives directly to the cathode slurry and coating it is simpler and currently more scalable than previously reported dry-coating methods[37]. The superior performance of KB-LGPS at higher current densities is attributed to the high Li-ion conductivity of LGPS, which facilitates efficient Li-ion transport and improves sulfur conversion efficiency, ensuring stable electrochemical kinetics under high-rate conditions.

The electrochemical impedance spectroscopy (EIS) analysis of the KB and KB + LGPS fresh coin cells revealed distinct differences in their charge transfer and ion transport behaviours [as shown in Fig. 4g and Table S1]. For the KB electrode, the equivalent circuit model (Fig. S5) comprised a series resistance ($R_1 = 1.39\ \Omega$), a charge transfer resistance ($R_2 = 91.51\ \Omega$) in parallel with a constant phase element ($CPE_1$), and a Warburg impedance ($W = 3.68\ \Omega$) representing diffusion limitations. In contrast, the KB + LGPS system required an additional parallel resistance ($R_3 = 1.07\ \Omega$) and $CPE3$ to account for the enhanced ionic pathways introduced by LGPS. The significantly lower charge transfer resistance ($R_2 = 43.7\ \Omega$) and Warburg coefficient ($W = 11.5\ \Omega$) in the KB + LGPS electrode indicated improved interfacial kinetics and reduced diffusion polarisation. The emergence of the low-impedance R3//CPE3 branch ($R_3 = 1.07\ \Omega$) strongly suggests that LGPS forms an independent solid-state $Li^+$ conduction network within the cathode, operating in parallel with the liquid electrolyte pathway. This dual-conduction mechanism aligns with the observed high-rate performance (800 mAh $g^{-1}$ at 1 C), where the LGPS-enabled pathway mitigates kinetic bottlenecks, enabling rapid $Li^+$ transport even under high current densities.

## Conclusion
A method using compatible solvents allowing for slurry coating for the preparation of a quasi-solid-state lithium–sulfur battery positive electrode

was developed, different solvents and binders suitable for LGPS were screened, and the coating process was optimised. The EIS and XRD results demonstrated that the LGPS did not react significantly with the EA and HB, and the less sharp and intense peaks suggest a loss of long-range order. However, some reactions did occur between LGPS and EA&HB. After being immersed and dried in EA&HB, the surface of the LGPS particles became rougher, the boundaries between particles became blurred, and new large particles or flat crystal plane structures appeared. These changes indicate that LGPS may have undergone partial dissolution during solvent treatment and experienced a recrystallisation process during drying, leading to a reorganisation of its microstructure. However, from an electrochemical performance perspective, these reactions are acceptable. The positive electrode, which also incorporates LGPS, has an average capacity of 1400 mAh $g^{-1}$ over multiple repeats and maintains a capacity of about 1040 mAh $g^{-1}$ after 200 cycles. The addition of LGPS altered the main electrode reaction, significantly increasing the reaction kinetics of conversion to short-chain polysulfides, which is otherwise the slowest step in the absence of LGPS and decreasing the electrode polarisation during the reduction of lithium polysulfide to $Li_2S$. The capacity of 800 mAh $g^{-1}$ was still maintained at high current rates (1 C), which is a significant improvement compared to the material without LGPS addition, which retains 400 mAh $g^{-1}$. After cycling at different rates and then cycling at a lower rate (0.05 C), there is still a 90% retention rate relative to the initial state. Combined with the electrochemical performance tests described above, it is confirmed that the addition of LGPS directly to the sulfur positive electrode indeed accelerates lithium-ion transport, reduces internal resistance, and significantly improves performance at high rates.

The results outlined here highlight the potential benefits of deploying Li-ion conducting components into Li-S electrodes. By improving the ionic conductivity of the electrodes, a significantly improved rate capability can be achieved, opening opportunities for the deployment of Li-S cells in power-sensitive applications whilst retaining the very high gravimetric energy density characteristics of Li-S. In summary, while significant progress has been made in the performance of quasi-solid-state lithium–sulfur batteries through the direct incorporation of LGPS in the cathode, especially in rate performance, many opportunities for improvement remain. Focusing on material stability, interface optimisation and, by extension, the use of pouch cells could pave the way for these cells to become a viable solution for high-energy, next-generation energy storage systems.

## Methods
### Preparation of quasi-solid-state lithium–sulfur batteries
LGPS (MSE Supplies, 99.9%) was immersed in different solvents: NMP (Sigma-Aldrich, 99%), Cyrene (Sigma-Aldrich, 98.5%), DMF (Sigma-Aldrich, 99.8%), DMSO (Sigma-Aldrich, 99.99%) and EA (Sigma-Aldrich, 99%) and HB (Sigma-Aldrich, 98%) to explore the reaction between them. PVDF-HFP (Sigma-Aldrich, 99%) was dissolved in EA at 60 °C, followed by gradual addition of HB at the same temperature in a volume ratio of 1:1 (EA:HB). Following this, the active materials: sulfur (Sigma-Aldrich, 99%), KB EC-600JD (Nouryon, 99%), PVDF-HFP in EA, SP (Fisher, 99%), LGPS (MSE supplies 99%) were mixed in a mass ratio of 50:25:10:5:10 under continuous stirring for 2 h in a small mixing pot by a magnetic stirrer (Camlab) at 60 °C. The control cell slurry was made using a similar method with S, KB, PVDF-HFP in EA, SP in a mass ratio of 55:30:10:5. After incorporating the active material and stirring, the coating was applied on carbon coated aluminium foil (MSE supplies) at a wet gap thickness of 30 μm and dried in a dry room at 60 °C for approximately 40 min. After drying, the electrodes were measured with a micrometre to be 27 μm in thickness, resulting in an areal capacity of 0.815 mAh $cm^{-2}$. The control cell electrodes had the same thickness with an areal capacity of 0.895 mAh $cm^{-2}$. Electrodes were then cut into circular discs (14 mm diameter) before being integrated into coin cells in a glove box using standard coin cell casings (MTI, Al-Clad CR2032 Coin Cell Case). For all cells, Celgard 2000, lithium foil (MTI, thickness: 0.6 mm, diameter: 16 mm, 99%), a single spring (MTI, Al-Clad stainless steel wave spring for CR2032/CR2016 Cases), and a single

**Article**

spacer (MTI, Stainless steel spacer for CR2032 cell 15.8×1.0 mm) were used. The electrolyte used was produced by dissolving lithium bis(trifluoromethanesulfonyl)imide (LiTFSI) (Sigma-Aldrich, 99%) in diethylene glycol dimethyl ether (G2) (Sigma-Aldrich, 99.9%) and 1,1,2,2-tetrafluoroethyl 2,2,2-trifluoropropyl ether (TTE) (Sigma-Aldrich, 99%), where the G2:LiTFSI (mol/mol) is 1.5:1, G2:TTE (vol/vol) is 1:1. Cells were then rested for 8 h prior to any testing to allow the electrolyte to fully percolate through the positive electrode.

### Characterisation and electrochemical measurements

To test the stability of LGPS in a range of common solvents, 1 g of the material was immersed in different solutions for 24 h before the solvent was fully evaporated. The samples were sealed with Kapton tape in the glove box to isolate them from the air. XRD (Stoe STADI-P) was then performed on the powder samples (sealed by Kapton tapes) using Mo Kα radiation ($\lambda = 0.0709$ nm), with data recorded in the 2-theta range of 2–40°. To conduct EIS, 0.5 g of dried solid was pressed manually into a small cell (diameter 0.8 mm, thickness 5 mm), and lithium chips were placed at both ends with EIS performed using frequencies from 1 MHz to 0.1 Hz using a Gamry potentiostat (Interface 1010E) with the cell housing shown in Fig. S6. The surface of the fresh positive electrode was also observed using a digital microscope (Keyence VHX-X1). Rate, cycling performance and EIS of the coin cell were obtained using a BioLogic battery test system (BCS-805, BioLogic). Cell cycling was performed between 1.5–3.5 V, with an initial cycle of 0.05 C (1 C = 1675 mA g$^{-1}$) conducted for all cells in advance of standard cycling at 0.1 C for the next. Rate performance testing was conducted with five cycles sequentially at each of 0.05 C, 0.1 C, 0.33 C and 1 C before returning to five further cycles at 0.05 C. Cyclic voltammetry was performed using a scan rate of 0.01 mV s$^{-1}$, between 1.0–4.0 V. Potentiostatic electrochemical impedance spectroscopy was conducted using a Gamry potentiostat (Interface 1010E) with a scan frequency of $0.1–1.0 \times 10^6$ Hz and a voltage perturbation of 5 mV. The EIS curves of coin cells were fitted using Relaxis. All scanning electron microscope imaging was conducted using a Zeiss Leo 1525 (Carl Zeiss AG, Jena, Germany), with the technique used to characterise the morphology of the prepared samples. Elemental mapping was conducted using an EDS detector on a Zeiss EVO MA 10. Analysis of the EDS measurements was conducted using INCA software. XCT was conducted by Zeiss Xradia Versa 620 (Carl Zeiss AG, Jena, Germany) with 1601 projections recorded over a 45 s exposure time with an accelerating voltage of 50 kV and a tube current of 90 μA. To achieve sufficient imaging resolution a 40× objective lens was used resulting in a 195.7 nm pixel size over a 392.29 μm field of view. Projections were reconstructed into image volumes using the filtered back projection algorithm with adaptive motion compensation, with all imaging undertaken using Avizo (V2020.2, Thermo-Fisher Scientific).

### Data availability

The data that supports the findings of this study are available from the authors on reasonable request; see author contributions for specific data sets.

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

## Acknowledgements
The authors would like to acknowledge the Faraday Institution (Faraday.ac.uk; EP/S003053/1) for funding via the LiSTAR research programme (FIRG083). JR and HL would like to acknowledge Innovate UK for funding (Project Number: 10040939). TM would like to acknowledge the Faraday Institute for funding via Nextrode (FIRG015, FIRG066). P.R.S. also acknowledges the Royal Academy of Engineering Chair in Emerging Technologies (CiET1718\59).

## Author contributions
B.P. proposed the research methodology, designed the experiments, prepared the samples, conducted all electrochemical measurements and material characterisation, analysed the results, and wrote the paper. H.L. participated in the research methodology, experimental design, and revised the paper. Y.G. and B.L. participated in data analysis and charting. F.L. recorded the scanning electron microscope (SEM) and energy-dispersive X-ray spectroscopy (EDS) images. H.C.W.P. performed the X-ray computed tomography (XCT) scans and image processing. L.R.B. participated in data processing and experimental design. T.S.M., P.R.S., R.J., and J.B.R. participated in research guidance, research methodology conception, and paper revision. All authors discussed the results and provided feedback.

## Competing interests
The authors declare no competing interests.
