## [Transparent Peer Review file · Communications Materials]

A High-Rate Lithium Sulfur Positive Electrode Using A Quasi-Solid-State Mechanism

Corresponding Author: Dr James Robinson

Version 0:

Decision Letter:

Dear Dr Robinson,

Thank you for submitting your manuscript "A High-Rate Lithium Sulfur Positive Electrode Using a Quasi-Solid-State Mechanism" to Communications Materials. It has now been seen by 2 referees, whose comments are appended below. You will see that while they find your work of interest, some important points are raised. We are interested in the possibility of publishing your study in Communications Materials, but would like to consider your response to these concerns in the form of a revised manuscript before we make a decision on publication.

In particular, Reviewer 2 suggests additional experimental evidence is needed. One significant concern is the increase in EIS semi-circle impedance after the combination of LGPS and EA/HB. This suggests that further experimental validation is needed to confirm the stability of the interaction between LGPS and EA/HB. Additionally, the small performance differences observed between the experimental and control samples should be thoroughly explained. The Reviewers also suggest logical inconsistencies, such as the discussion of binder dispersion in electrodes, which is not directly related to LGPS's role in enhancing ion transport.

We therefore invite you to revise and resubmit your manuscript, taking into account the points raised.

When submitting your revised manuscript, please include the following:

-A response letter with a point-by-point reply to each of the referee comments and a description of changes made. Please include the complete referee report in the response letter. Please note that the response letter must be separate to the cover letter to the editors.

-A marked-up version of the manuscript with all changes to the text in a different colored font. Please do not include tracked changes or comments. Please select the file type 'Revised Manuscript - Marked Up' when uploading the manuscript file to our online system.

-A clean version of the manuscript. Please select the file type 'Article File'.

-An updated [Editorial Policy](https://www.nature.com/documents/nr-editorial-policy-checklist.zip) checklist, uploaded as a 'Related Manuscript File' type. This checklist is to ensure your paper complies with all relevant editorial policies. If needed, please revise your manuscript in response to these points. Please note that this form is a dynamic 'smart pdf' and must therefore be downloaded and completed in Adobe Reader. Clicking this link will download a zip file containing the pdf.

In the event that your manuscript is accepted we will provide detailed guidance on our journal policies and formatting. You may however wish to ensure that the manuscript complies with our house style at this stage. See our style and formatting

guide (<https://www.nature.com/documents/commsj-phys-style-formatting-guide-accept.pdf>) and checklist (<https://www.nature.com/documents/commsj-phys-style-formatting-checklist-article.pdf>) for reference.

Data availability statements and data citations policy: All Communications Materials manuscripts must include a section titled "Data Availability" at the end of the Methods section or main text (if no Methods). More information on this policy, and a list of examples, is available at <http://www.nature.com/authors/policies/data/data-availability-statements-data-citations.pdf>.

- Accession codes for deposited data
- Other unique identifiers (such as DOIs and hyperlinks for any other datasets)
- At a minimum, a statement confirming that all relevant data are available from the authors
- If applicable, a statement regarding data available with restrictions
- If a dataset has a Digital Object Identifier (DOI) as its unique identifier, we strongly encourage including this in the Reference list and citing the dataset in the Data Availability Statement.

DATA SOURCES: We strongly encourage authors to deposit all new data associated with the paper in a persistent repository where they can be freely and enduringly accessed. We recommend submitting the data to discipline-specific, community-recognized repositories, where possible and a list of recommended repositories is provided at <http://www.nature.com/sdata/policies/repositories>.

If a community resource is unavailable, data can be submitted to generalist repositories such as [figshare](https://figshare.com/) or [Dryad Digital Repository](http://datadryad.org/). Please provide a unique identifier for the data (for example a DOI or a permanent URL) in the data availability statement, if possible. If the repository does not provide identifiers, we encourage authors to supply the search terms that will return the data. For data that have been obtained from publically available sources, please provide a URL and the specific data product name in the data availability statement. Data with a DOI should be further cited in the methods reference section.

Please use the following link to submit your documents:

Link Redacted

We hope to receive your revised paper within three months; please let us know if you aren't able to submit it within this time so that we can discuss how best to proceed. If we don't hear from you, and the revision process takes significantly longer, we will close your file. In this event, we will still be happy to reconsider your paper at a later date, as long as nothing similar has been accepted for publication at Communications Materials or published elsewhere in the meantime.

Please do not hesitate to contact me if you have any questions or would like to discuss these revisions further. We look forward to seeing the revised manuscript and thank you for the opportunity to review your work.

Best regards,

Guangmin Zhou, PhD
Editorial Board Member
Communications Materials
orcid.org/0000-0002-3629-5686

Reviewers' comments:

Reviewer #1 (Remarks to the Author):

The authors developed a new concept of incorporating sulfide solid electrolyte into the sulfur cathodes of Li-S batteries. The influence of different solvent systems and binders on the electrochemical performance of the positive electrode was investigated. The authors find that the addition of LGPS improved lithium-ion transport, reduced internal resistance, and enhanced reaction kinetics. Higher capacity and longer cycle life was achieved. I like the concept of quasi-solid-state Li-S cells as it offers great insight for a differently functioned Li-S cells. I would like to recommend for publications pending some comments below:

1. The title is misleading. Quasi-solid-state mechanism often means the mechanism of the reaction. The authors meant to be a quasi-solid-state configuration
2. The Nyquist plot in Figure 2b is noisy and may not be meaningful. The authors need to re-do it or explain and justify its correctness.

3. In figure c, the cell of KB+LGPS eventually fades to be the same capacity with KB. Why is this happening?

Reviewer #2 (Remarks to the Author):

The authors have designed and explored the possible implementation of LGPS solid electrolytes as catholytes in Li-S batteries. They have presented that such integration optimizes the Li-ion conducting pathway and thereby enhance rate capability and suppress sulfur shuttling to extend the cycle life. However, the study lacks experimental evidence to support their theory. Also, experimental results are questionable and are thought to need further optimization. In conclusion, this research is not yet ready to be published and needs further investigations.

1. Although the authors suggest stable interaction between LGPS and EA/HB solvent, significant change in the semi-circles (resistance) is observable in Figures 2a and 2c. How can this be explained? Further characterization is thought to be necessary.

2. It is hard to determine the degree of dispersion of binders from SEM images presented in Figure 3. Also, although dispersion is important in all wet-processed electrodes, it is thought to be irrelevant to the mechanisms behind the optimization of Li-ion conducting pathways and the suppression of shuttling effects. Likewise, it is advised that Figure 4 also be adjusted.

3. The two samples shown in Figure 5a shows trivial difference in overpotential throughout the charge-discharge voltage curves. One could suggest that if the high conductivity of LGPS was to support ion transport, the sample with KB+LGPS should exhibit smaller overpotential. Could the authors elaborate? And even so, the two samples exhibit different amounts of capacities. Why?

4. The discharge capacities shown in Figure 5c and 5d don't match, hence any results driven are thought to be very misleading. Further optimization of experimental protocols is suggested.

5. The results that the authors drew from the EIS data in Figure 5g are questionable. The relation between the difference in the Warburg element and the promotion of diffusion should be further elaborated, as the explanations the authors have provided are vague and confusing. Also, the relationship between the promotion of diffusion and the rate capability should be explained. The good-fit achieved by adding a parallel resistor is not sufficient to suggest that there is a different conduction pathway. Further investigation to support the idea should be conducted.

Communications Materials is committed to improving transparency in authorship. As part of our efforts in this direction, we are now requesting that all authors identified as 'corresponding author' create and link their Open Researcher and Contributor Identifier (ORCID) with their account on the Manuscript Tracking System prior to acceptance. ORCID helps the scientific community achieve unambiguous attribution of all scholarly contributions. You can create and link your ORCID from the home page of the Manuscript Tracking System by clicking on 'Modify my Springer Nature account' and following the instructions in the link below. Please also inform all co-authors that they can add their ORCIDs to their accounts and that they must do so prior to acceptance.

Version 1:

Decision Letter:

Dear Dr Robinson,

Thank you for submitting your manuscript, "A High-Rate Lithium Sulfur Positive Electrode Using A Quasi-Solid-State Mechanism", to Communications Materials. It has now been seen again by 2 referees, whose comments are appended below. You will see that while Reviewer 1 thinks your work is acceptable, Reviewer 2 has raised substantial concerns that must be addressed. In light of these comments, we cannot accept the manuscript for publication, but are interested in

considering a revised version that addresses these serious concerns.

In particular, they mention that the evidence for LGPS is not convincingly supported, which may require further experiments.

We hope you will find the referees' comments useful as you decide how to proceed. Should further experimental data or analysis allow you to address these criticisms, we would be happy to look at a substantially revised manuscript. However, please bear in mind that we will be reluctant to approach the referees again in the absence of major revisions. If the revision process takes significantly longer than twelve weeks, we will be happy to reconsider your paper at a later date, as long as nothing similar has been accepted for publication at Communications Materials or published elsewhere in the meantime.

When submitting your revised manuscript, please include the following:

-A response letter with a point-by-point reply to each of the referee comments and a description of changes made. Please include the complete referee report in the response letter. Please note that the response letter must be separate to the cover letter to the editors.

-A marked-up version of the manuscript with all changes to the text in a different colored font. Please do not include tracked changes or comments. Please select the file type 'Revised Manuscript - Marked Up' when uploading the manuscript file to our online system.

-A clean version of the manuscript. Please select the file type 'Article File'.

Please ensure that the following requirements are met, and that any relevant checklists are completed and uploaded under the 'Related Manuscript file' type with the revised article.

Please use the following link to submit your revised manuscript files:

Link Redacted

Please do not hesitate to contact me if you have any questions or would like to discuss the required revisions further. Thank you for the opportunity to review your work.

Best regards,

Jet-Sing Lee, PhD
Senior Editor
Communications Materials

Reviewers' comments:

Reviewer #1 (Remarks to the Author):

The authors have answered my queries well.

Reviewer #2 (Remarks to the Author):

Although the authors have made efforts to address my previous comments, the revisions provided do not adequately meet scientific standards.

Comment 1: The authors claim the superiority of LGPS with EA&HB by comparing it to LGPS with NMP. However, the resistance of LGPS with EA&HB is significantly higher than that of LGPS without a solvent. This raises concerns about the purported enhancement of the ion-conductive pathway of LGPS within the electrodes.

Comment 2: The data presented in Figure 4 continues to raise questions regarding the distribution of LGPS. The aggregation of PVDF might impede the proper distribution of LGPS. Additionally, the EDS peaks for Ge and P are too weak to convincingly demonstrate well-distributed LGPS.

Comment 3: The authors acknowledge substantial performance variability between cells, even with identical electrodes. However, the observed voltage differences of 50-100 mV in one cell per sample cannot be attributed solely to the addition of LGPS.

Comment 4: Despite modifications to the rate-capability data, the overall cycling and rate-capability performances do not convincingly demonstrate the effect of LGPS.

In conclusion, I believe this research is not yet suitable for publication in its current form and recommend considering submission to a more specialized journal.

Communications Materials is committed to improving transparency in authorship. As part of our efforts in this direction, we are now requesting that all authors identified as 'corresponding author' create and link their Open Researcher and Contributor Identifier (ORCID) with their account on the Manuscript Tracking System prior to acceptance. ORCID helps the scientific community achieve unambiguous attribution of all scholarly contributions. You can create and link your ORCID from the home page of the Manuscript Tracking System by clicking on 'Modify my Springer Nature account' and following the instructions in the link below. Please also inform all co-authors that they can add their ORCIDs to their accounts and that they must do so prior to acceptance.

Version 2:

Decision Letter:

Dear Dr Robinson,

Your manuscript titled "A High-Rate Lithium Sulfur Positive Electrode Using A Quasi-Solid-State Mechanism" has now been seen again by our referees, whose comments appear below. In light of their advice I am delighted to say that we are happy, in principle, to publish a suitably revised version in Communications Materials.

We therefore invite you to edit your manuscript to comply with our journal policies and formatting style in order to maximise the accessibility and therefore the impact of your work.

EDITORIAL REQUESTS

* Your manuscript should comply with our policies and format requirements, detailed in our style and formatting guide (<https://www.nature.com/documents/commsj-phys-style-formatting-guide-accept.pdf>).

* Please edit your manuscript according to the editorial requests in the attached table, and outline revisions made in the right hand column. If you have any questions or concerns about any of our requests, please do not hesitate to contact me. It is important that each request be addressed in order to avoid delays in accepting your manuscript. Please upload the completed table with your manuscript files as a Related Manuscript file.

* The editorial requests table also includes a full list of the files that must be provided upon resubmission. Please upload your files according to this table.

OPEN ACCESS

Communications Materials is a fully open access journal. Articles are made freely accessible on publication. For further information about article processing charges, open access funding, and advice and support from Nature Research, please visit <https://www.nature.com/commsmat/open-access>

Please use the following link to submit your revised files:

Link Redacted

We hope to hear from you within two weeks; please let us know if the process may take longer.

Best regards,

Dr Jet-Sing Lee
Senior Editor
Communications Materials

REVIEWERS' COMMENTS:

Reviewer #2 (Remarks to the Author):

All the concerns are adequately addressed. I would like to recommend acceptance of this version.

We would like to thank the reviewers for their time and effort in providing comments on our article. Having reviewed these comments, we have amended the submission based on their insights. The changes to the article are highlighted in the resubmitted manuscript (additions in green, deletions in red) with responses to the reviewers' comments below on a point-by-point basis.

Reviewer 1

The authors developed a new concept of incorporating sulfide solid electrolyte into the sulfur cathodes of Li-S batteries. The influence of different solvent systems and binders on the electrochemical performance of the positive electrode was investigated. The authors find that the addition of LGPS improved lithium-ion transport, reduced internal resistance, and enhanced reaction kinetics. Higher capacity and longer cycle life was achieved. I like the concept of quasi-solid-state Li-S cells as it offers great insight for a differently functioned Li-S cells. I would like to recommend for publications pending some comments below:

The title is misleading. Quasi-solid-state mechanism often means the mechanism of the reaction. The authors meant to be a quasi-solid-state configuration

Naturally, we had no intention to mislead readers on the content of our manuscript and agree that the QSS mechanism typically refers to the reaction mechanism. Our intention had been to emphasise that this cell electrode is intended to operate in a different electrolyte system, and by extension via a different mechanism to more conventional Li-S systems which are commonly reported i.e. 1M LiTFSI in DOL:DME with XM LiNO₃. To avoid confusion, we have changed the to "A Quasi-Solid-State High-Rate Lithium Sulfur Positive Electrode incorporating Li₁₀GeP₂S₁₂" which highlights the primary innovation of this study.

The Nyquist plot in Figure 2b is noisy and may not be meaningful. The authors need to re-do it or explain and justify its correctness.

A less noisy Nyquist plot of LGPS+NMP has been redone. The new Nyquist plot has clearer arcs in the high frequency region.

In Figure c, the cell of KB+LGPS eventually fades to be the same capacity with KB. Why is this happening?

As can be seen in Figure 5c the performance benefits offered by the inclusion of LGPS into the electrolyte manifest for ca. 80 cycles before the performance converges to the electrode manufactured without LGPS. At present we hypothesis this is likely due to the relatively wide potential window which we have used in this study, which we believe may be too wide for the current system and result in reactions occurring between the electrolyte (and degradation products thereof) and the LGPS. We are actively investigating strategies to improve this degradation rate and aim to publish this in future studies which will include ongoing

electrochemical analysis and characterisation in a range of different electrolytes; however consider this to be out of scope for this study due to the volume of work required and the size of the resultant manuscript.

Reviewer 2

The authors have designed and explored the possible implementation of LGPS solid electrolytes as catholytes in Li-S batteries. They have presented that such integration optimizes the Li-ion conducting pathway and thereby enhance rate capability and suppress sulfur shuttling to extend the cycle life. However, the study lacks experimental evidence to support their theory. Also, experimental results are questionable and are thought to need further optimization. In conclusion, this research is not yet ready to be published and needs further investigations.

Although the authors suggest stable interaction between LGPS and EA/HB solvent, significant change in the semi-circles (resistance) is observable in Figures 2a and 2c. How can this be explained? Further characterization is thought to be necessary

Compared to the commonly used NMP, the impedance of the EA- and HB-treated LGPS changes less. the impedance of the NMP-treated LGPS undergoes an improvement of several orders of magnitude.

The XRD performed showed no significant change in the position of the peaks of LGPS after solvent treatment and drying, indicating that the LGPS composition did not undergo significant changes. However, the reduced peak intensity suggests that there may be a process of dissolution then recrystallization of LGPS crystals which is likely to change the impedance characteristics of the material.

Therefore, we do not believe that EA&HB did not react with LGPS at all, however it is clearly a more viable solvent than NMP and we believe the changes in impedance are within acceptable limits. The reaction of solid electrolytes and solvents for this form of electrode has not been widely studied or discussed (with dry processing used in the previous example highlighted in text), and the available studies do not fully expose the reaction mechanism of LGPS with various solvents.

The observation of LGPS particle morphology using SEM/TEM to confirm the occurrence of particle size changes or agglomeration may help to explain in detail the reaction of LGPS with solvents, but SEM/TEM scans are not possible to perform under current laboratory conditions due to the air sensitivity of LGPS.

We understand and acknowledge the reviewers concern in this regard and have ensured this section of the manuscript has been revised with more precise language to avoid any misreading

of our hypothesis.

It is hard to determine the degree of dispersion of binders from SEM images presented in Figure 3. Also, although dispersion is important in all wet-processed electrodes, it is thought to be irrelevant to the mechanisms behind the optimization of Li-ion conducting pathways and the suppression of shuttling effects. Likewise, it is advised that Figure 4 also be adjusted.

The voids of C, S and F shown on EDS in Figure 4 may be due to the agglomerates or jelly-like colloids of PVDF-HFP which disappeared after drying. It is one of the disadvantages PVDF-HFP as this part discussed. The dispersion of binders is less irrelevant to shutting effects. The well dispersed Ge and P showed LGPS was uniformly distributed which would provide consistent Li-ion conducting pathways. This part of the article has been revised with more details.

The two samples shown in Figure 5a shows trivial difference in overpotential throughout the charge-discharge voltage curves. One could suggest that if the high conductivity of LGPS was to support ion transport, the sample with KB+LGPS should exhibit smaller overpotential. Could the authors elaborate? And even so, the two samples exhibit different amounts of capacities. Why?

We agree with the reviewer and this result can be seen in Figure 5b where there was a significant change in the difference between the oxidation and reduction peaks. Here the addition of LGPS reduces the difference from 410 mV to 360 mV in the first reduction peak and 730 mV to 600 mV in the second reduction peak. This difference is admittedly not as clear in the full cell cycling; however, there are clear variations in the discharge curve particularly around the plateau transition which is steeper for the cell which does not contain LGPS. We believe there are two possible reasons for the increase in capacity the first is that the LGPS itself contains S, and therefore may undergo some side reactions; however, given the relatively small addition of LGPS to the electrode we do not think this is a likely explanation for the full difference. Indeed, we suspect that the change in gradient at ca. 1.6 V may be assignable to these side reactions with the electrolyte (akin to the electrochemical decomposition of LiNO_3 below 1.8 V often seen in more conventional Li-S cells). The bulk of this difference, we believe, is more likely due to an improved sulfur utilisation which the addition of the LGPS enables.

The discharge capacities shown in Figure 5c and 5d don't match, hence any results driven are thought to be very misleading. Further optimization of experimental protocols is suggested.

Since each coin cell is assembled by hand, there is a natural variation of in capacity which can arise. To ensure the validity of the results we have rerun the rate performance test on a newly manufactured electrode (prepared following the reviewers comments) and replaced Figure 5d.

As seen the results are representative of the previous rate performance test with significant improvements in performance across all rates when LGPS is incorporated, in particular at 1C.

The results that the authors drew from the EIS data in Figure 5g are questionable. The relation between the difference in the Warburg element and the promotion of diffusion should be further elaborated, as the explanations the authors have provided are vague and confusing. Also, the relationship between the promotion of diffusion and the rate capability should be explained. The good-fit achieved by adding a parallel resistor is not sufficient to suggest that there is a different conduction pathway. Further investigation to support the idea should be conducted.

We refitted the equivalent circuit model. Where the KB system is modelled as R1 (electrolyte impedance) - [R2//CPE1] (liquid path) - [CPE2 + Warburg] (diffusion impedance). The KB+LGPS system is modelled as R1 (electrolyte impedance) - [R2//CPE1] (liquid path) - [R3//CPE3] (LGPS solid path) - [CPE2 + Warburg] (diffusion impedance)

Name	Resis tance 1	Resist ance 2	CPE Q 1	CPE Alpha 1	Resista nce 3	CPE Q 2	CPE Alpha 2	CPE 3	Warbu rg Coeff 1
KB	41.8 68 (0.27 %)	2745. 3 (0.36 %)	1.01E- 5 (1%)	0.7647 4 (0.18 %)	N.A.	0.0237 28 (7.2 %)	1 (1.2 %)	N.A.	110.8 (6.3%)
KB+LGPS	59.9 86 (2%)	1318. 7 (0.46 %)	1.049E -5 (0.8%)	0.7531 4 (0.25 %)	32.143 (4%)	0.0236 48 (13%)	1 (6.4%)	0.0097 08 (7.1%)	11.516 S(11%)

Based on these models the addition of LGPS reduced the charge transfer impedance (R_{ct}) from 2745.3 Ω to 1318.7 Ω by about 52%, suggesting that LGPS improves the ion transfer efficiency at the electrode/electrolyte interface, consistent with its high ionic conductivity. The KB+LGPS

system adds $R_3 = 32.143 \Omega$. This low impedance path can be attributed to the solid-state ionic conduction network formed by the LGPS in the cathode, which is connected in parallel with the liquid electrolyte path (R_2), providing an additional lithium-ion transport channel. The Warburg coefficient is reduced by an order of magnitude from 110.8Ω to 11.516Ω , suggesting that the LGPS significantly improves lithium-ion diffusion through the solid-state conduction network significantly improves the lithium ion diffusion kinetics and reduces the diffusion polarisation. The addition of the $R_3//CPE_3$ element indicates the presence of two parallel Li-ion transport paths in the system: a liquid electrolyte path ($R_2 = 1318.7 \Omega$): dominating the conventional ion transport, and an LGPS solid-state path ($R_3 = 32.143 \Omega$): providing a low-impedance supplementary channel.

In summary, the impedance of R_3 (32.143Ω) is much lower than the R_{ct} of the liquid pathway (1318.7Ω), suggesting that the LGPS provides a more efficient lithium-ion transport channel through the solid-state network. The significant decrease in R_{ct} (52%) directly supports the improvement in the interfacial dynamics, which is consistent with the experimental enhancement in capacity retention to 800 mAh g^{-1} at $1C$ (KB+LGPS). The decrease in Warburg coefficient suggests that the solid-state network of LGPS bypasses the diffusion bottleneck of the liquid electrolyte and directly accelerates Li-ion transport in the anode.

The relevant analyses have been modified in the original manuscript.

We aim to further explore the performance of this electrode type with a range including GITT, and we fully agree with its value for in-depth analysis of electrode reaction kinetics. However, due to the current experimental conditions and project cycle, we are unable to complete the GITT test in the revised stage for the time being. This work will be included in a wider study focussed investigating the wider QSS mechanism and the incorporation of LGPS and other conductive components; however given the need to adequately explain all aspects of this research, and the prolonged testing which will be required we do not feel it is within scope of the study presented.

We would like to thank the reviewer for their further efforts in commenting on the work and have made efforts to ensure that these comments are addressed to enable publication of the work as detailed below.

Comment 1: The authors claim the superiority of LGPS with EA&HB by comparing it to LGPS with NMP. However, the resistance of LGPS with EA&HB is significantly higher than that of LGPS without a solvent. This raises concerns about the purported enhancement of the ion-conductive pathway of LGPS within the electrodes.

We acknowledge there is an increase in the impedance of LGPS after EA&HB treatment and have hypothesised this is due to a partial recrystallisation during electrode manufacture resulting in a change to particle morphology. The X-ray diffraction results indicate that the main characteristic peaks of LGPS have not disappeared or shifted with weakened peak intensity may be attributed to the hypothesised dissolution and recrystallisation process. We have confirmed that there is indeed a change in the particle morphology by our supplementary SEM testing of LGPS before and after solvent treatment which are shown in Figure 2. LGPS treated with the solvent (Figures 2f and i) exhibits rounder, rougher particle surfaces, blurred particle boundaries, and the emergence of new large particles or flat crystal face structures. These changes suggest that LGPS may have undergone partial dissolution during solvent treatment and recrystallisation during drying, leading to the reorganisation of its microstructure. We acknowledge that EA&HB still influences the properties of LGPS, but compared to other solvents we have tested, such as NMP, EA&HB has a relatively minor impact. There may be more suitable solvent/binder systems, which is one of our future research directions, and indeed dry coating of this electrode may be a viable solution to minimise this effect; however this is not without its challenges and given we have attempted to maintain compatibility with standardised manufacturing routes EA&HB remains the most suitable candidate we have screened to date.

The revised paragraphs and figures are highlighted in yellow below:

Figure 2 Electrochemical impedance spectroscopy (a-c), X-ray diffraction (d) and scanning electron microscope (e-j) of LGPS with different solvents: a), e) and h) LGPS; b), f) and i) LGPS with EA&HB; c), g) and j) LGPS with NMP; d) XRD of LGPS, LGPS+ EA&HB and LGPS reference peaks.

The absence of significant peak displacement and the appearance of new peaks indicate that LGPS did not undergo phase transformation or decomposition into polysulfides or other crystalline compounds such as Li_4GeS_4 , after EA and HB treatments. The decrease in peak intensity may result from a reduction in crystallinity, indicating that the LGPS particles become more disordered after solvent treatment while preserving their original crystalline phase. It is

possible that there is a process of dissolution and recrystallisation. Moreover, during the solvent treatment of LGPS, a slurry-like morphology was observed with the naked eye, which stuck to the bottom of the container like scale after evaporation.

To verify that LGPS undergoes a dissolution-recrystallisation process after EA&HB treatment, SEM was used to observe the microstructure as shown in Figure 2(e, f, h, i). The original LGPS [Figure 2(e, h)] has smaller, sharper particles with distinct edges, exhibiting typical crystalline particle aggregation characteristics. After EA&HB treatment, the LGPS particles become rounder and blunter overall, with a rough surface and blurred boundaries between particles, as shown in Figure 2(f). At lower magnification in Figure 2(i), large flat areas and crystalline-like structures are clearly visible, which may result from the regrowth and reorganisation of crystals. Meanwhile, the morphology of LGPS after NMP treatment [Figure 2(g,j)] is completely different, having lost its original crystalline structure.

Comment 2: The data presented in Figure 4 continues to raise questions regarding the distribution of LGPS. The aggregation of PVDF might impede the proper distribution of LGPS. Additionally, the EDS peaks for Ge and P are too weak to convincingly demonstrate well-distributed LGPS.

We agree with the reviewer that there are likely routes to further optimise the coating process for this EA&HB/PVDF-HFP solvent-binder system. Most pressingly there are some cracks in regions of the positive electrode over a full coating although our supplementary SEM (Figure 3e) shows that the PVDF-HFP binder is present in the cracks and does not appear to be agglomerated, forming a strong microstructural network. Furthermore, the additional XCT images now included (Figures 3h and i) show a well dispersed sulfur electrode which is similar to what would be expected. It should be noted that the CT does not provide sufficient resolution to observe the LGPS particles within the system. Whilst the Ge and P peaks are comparatively weak, they are in proportion to the magnitude expected based on the relatively small amount of LGPS incorporated in the electrode. The well dispersed nature of this signal also indicates the presence of Ge and P across the surface imaged which could not be explained by any other component within this system. Given the observed distribution of the sulfur and PVDF alongside the energy dispersive X-ray spectroscopy results shown and the improved performance obtained electrochemically we see no reason to believe that the LGPS is not reasonably well dispersed within the electrode.

The revised paragraphs and figures are highlighted in yellow below:

Figure 3 Scanning electron microscope (a-e), optical microscopy measurements (f, g) and X-ray computed tomography (h, i) of the positive electrode incorporating sulfur, LGPS, ketjen black, and PVDF-HFP binder. a) KB powder; b) a side profile of the electrode highlighting the strong adhesion to the Al foil and dispersion of sulfur in the electrode; c) the binder connectivity between the particles in the electrode; d) KB with uniform particles in the electrode; e) PVDF-HFP between cracks; f) surface topography of the electrode obtained using optical microscopy; g) cross-surface profile of the electrode highlighting the uniform thickness observed following manufacture; h) morphology of electrode; i) morphology of S distribution within electrode. Sulfur particles may be dispersed in the KB network as shown in Figure 3(h,i). Many pores can also be seen, which provide space for the volume change of sulfur during charging and

discharging processes. Although cracks are inevitable in the cathode, Figure 3 (e) shows the binder between the cracks, which stabilises the structure while also providing a good conductive network. The digital microscope image [Figure 3(f,g)] shows a uniform thickness across the surface of the positive electrode.

As the reviewers pointed out, another concern raised was the agglomeration of the binder. As shown in the mapping of the binder element F in Figure 4i, there are some densely concentrated bright spots, indicating that binder agglomeration may exist in the cathode. It was also observed in Figures 4g and 4h that there are dense regions of F and the presence of C, but these are complementary to the presence of S. This indicates that the aggregation in the figure is primarily composed of C and binder, with no presence of S. However, the distribution of Ge and P appears unrelated to the dense occurrence of F, suggesting that binder aggregation does not affect the distribution of LGPS throughout the cathode.

There are many factors that influence the intensity of EDS peaks, such as atomic concentration (At%), fluorescence yield (ω), detector efficiency (η), and ionisation cross-section (σ). Among these, atomic concentration plays a decisive role, and peak intensity is directly proportional to atomic concentration. In the positive electrode material, the mass ratio of S:Li₁₀GeP₂S₁₂ is 55:10. The atomic concentration ratio is S:Ge:P = 89:0.7:1.5. This trend is consistent with the intensity ratio of the peaks for S, Ge, and P. Additionally, EDS peaks are point analyses: the intensity of a single peak only reflects the local composition and cannot determine overall uniformity. The uniform distribution of LGPS is confirmed by mappings of Ge and P.

Figure 4 e) EDS raw image of KB+LGPS positive electrode; f) EDS analysis of KB+LGPS positive electrode; g) EDS mapping image of C of KB+LGPS positive electrode; h) EDS mapping image of S of KB+LGPS positive electrode; i) EDS mapping image of F of KB+LGPS positive electrode; j) EDS mapping image of Ge of KB+LGPS positive electrode; k EDS mapping image of P of KB+LGPS positive electrode.

Comment 3: The authors acknowledge substantial performance variability between cells, even with identical electrodes. However, the observed voltage differences of 50-100 mV in one cell per sample cannot be attributed solely to the addition of LGPS.

We agree with the reviewer that cell-to-cell variability is a common challenge in Li-S batteries. A reduction of 50–100 mV does not seem very significant, but thanks to our excellent G2-TTE electrolyte, our battery has a relatively low overpotential of approximately 250 mV. Based on this, there is not much room for improvement.

In addition, our tests were conducted at lower magnifications and lower scan speeds. Therefore, we supplemented the charge-discharge curves (Figure 4a) at different rates (1C and 0.01C),

which clearly show the excellent performance of LGPS in reducing overpotential at high rates. This was repeated for three individual cells for each of the electrodes (S/KB and S/KB/LGPS) with statistical variation now included in the Figure showing high confidence that the variation in performance is indeed provided by the inclusion of LGPS, which is the only difference between the cells.

Combining CV, rate, long cycle, EIS, and circuit fitting analyses, we have reason to believe that directly adding LGPS to the cathode is an effective way to improve the performance of Li-S batteries.

The revised paragraphs and figures are highlighted in yellow below:

Figure 5. Electrochemical performance of KB and KB-LGPS : during a) charge and discharge at 0.01C and 1.0C; b) 1st cycle CV at 0.01 mV/s; c) long term cycle performance the first cycle at 0.05C then at 0.1C; d) mean rate performance with standard deviation shown in shadow and KB+LGPS discharge and charge curve; e) EIS of fresh coin cells. Figure 4(a) shows the charge-discharge curves of the two groups at 0.01C and 1C rates. The charge-discharge curves without LGPS showed overpotentials of 217 mV and 855 mV at 0.01C and 1C, respectively. However, the charge-discharge curves with LGPS showed significantly lower overpotentials of 116 mV (0.01C) and 464 mV (1C). This demonstrates the enormous potential of LGPS in reducing overpotential.

Comment 4: Despite modifications to the rate-capability data, the overall cycling and rate-capability performances do not convincingly demonstrate the effect of LGPS.

We appreciate the reviewer's concern about substantial performance variability between cells. We agree that more battery data needs to be included in order to draw more accurate conclusions. Considering the uniqueness of each battery, we tested three individual cells for each electrode to determine their rate performance and organised the data into averages and standard deviations, which are shown in Figure 5d. These data strongly prove the excellent rate performance of batteries with added LGPS. Compared with batteries without added LGPS, the capacity at 1C increased from around 400 to around 800 mAh/g.

Figure 5. Electrochemical performance of KB and KB-LGPS : d) mean rate performance with standard deviation shown in shadow and KB+LGPS discharge and charge curve;